# Boosting-Inspired Validation of Retrieval-Augmented Generation in Structured Scientific Knowledge Bases

## Abstract

Large Language Models (LLMs) enhanced with Retrieval-Augmented Generation (RAG) achieve remarkable results, yet they often hallucinate or provide incomplete answers. This poses critical challenges in scientific knowledge domains where factuality and precision are essential. In this paper, we propose a boosting-inspired evaluation framework for RAG that combines iterative error reduction with forward-looking retrieval mechanisms from FLARE. Unlike existing work that primarily optimizes retrieval or ranking, our focus is on the validation loop itself. We validate the framework in a controlled scenario using Citavi, a structured literature management system, serving as a reproducible environment for testing. Results indicate that strict substring matching underestimates semantic correctness, while boosting-inspired metrics highlight when expansion is necessary. This proof-of-concept demonstrates technical feasibility and motivates iterative, semantic validation for future scientific assistants.

## *Human Foreword*

This paper is a meta-experiment. It was created in a continuous collaboration where a human researcher acted as supervisor and generative AI systems (ChatGPT-4 and GPT-5) took on the role of creator of the scientific proof-of-concept. The entire workflow - from exploring the research question, through drafting code and structuring the validation scenario, to producing the manuscript - was conducted within a single ChatGPT chat.

To ensure transparency, the GitHub repository referenced in the acknowledgements provides open access to the complete chat history (in German and English translation), the validation project, and the Citavi test project.

## 1   Introduction

Large Language Models (LLMs) have rapidly advanced natural language processing and are increasingly applied in scientific and industrial domains. Despite their remarkable capabilities, a persistent challenge remains: LLMs tend to hallucinate, producing factually incorrect or unverifiable content [1]. This shortcoming is particularly problematic in scientific knowledge bases, where accuracy, reproducibility, and transparency are essential.

Retrieval-Augmented Generation (RAG) [2] mitigates this issue by combining parametric memory stored in model weights with non-parametric memory retrieved from external corpora. While RAG improves factual grounding, it lacks systematic validation loops to ensure that retrieved evidence is sufficient and that generated answers remain reliable. In practice, validation is often reduced to ranking metrics, leaving gaps in coverage and robustness unaddressed.

Boosting methods, such as Gradient Boosting [3] and XGBoost [4], demonstrate the effectiveness of iteratively reducing residual errors. Similarly, FLARE [5] introduced forward-looking retrieval, in which intermediate predictions guide expansion towards missing evidence. Both approaches highlight the importance of iterative refinement, a principle not yet fully leveraged in RAG validation.

This paper is motivated by the need for reliable validation mechanisms in knowledge-intensive environments. We therefore introduce a methodology that integrates boosting-inspired residual tracking with FLARE-style expansion, enabling a dedicated evaluator for RAG. To examine its feasibility, we conduct a pilot validation in a structured environment (Citavi), which provides citations, abstracts, and hierarchical knowledge suitable for controlled testing. The results reveal the limitations of strict string-matching metrics and highlight the necessity of semantic evaluation for future iterations. Together, these contributions demonstrate the potential of boosting-inspired validation as a new direction for improving the robustness of retrieval-augmented generation in scientific knowledge bases.

## 2 Related Work

Our work builds on three main strands of research: ensemble learning and boosting, retrieval-augmented models, and evaluation of hallucinations and factuality. Each area contributes important foundations, yet none addresses the specific problem of designing validation loops for Retrieval-Augmented Generation (RAG).

### 2.1 Boosting and Ensembles

Ensemble learning, and boosting in particular, has proven to be a powerful method for iterative error reduction. Friedman introduced Gradient Boosting [3], which was later extended in practical implementations such as XGBoost [4]. Further theoretical contributions, such as the comprehensive review by Bühlmann and Hothorn [6], and the classic textbook by Hastie, Tibshirani, and Friedman [7], emphasize the principle of repeatedly fitting residuals to improve predictive performance. This principle inspires our evaluator design, which aims to detect and act upon coverage gaps in retrieved evidence.

### 2.2 Retrieval-Augmented Models

In parallel, retrieval-augmented models have become central to modern language technologies. Lewis et al. presented RAG [2], combining parametric knowledge embedded in model weights with non-parametric retrieval. Guu et al. extended this line with REALM [8], where retrieval is interleaved during pretraining, and Karpukhin et al. introduced Dense Passage Retrieval (DPR) [9]. Izacard and Grave proposed Fusion-in-Decoder (FiD) [10], while Borgeaud et al. scaled retrieval to trillions of tokens in RETRO [11]. More recently, Izacard et al. introduced FLARE [5], which uses forward-looking predictions to actively expand retrieval. Dialogue-focused systems [12] and domain-specific adaptations such as scientific question answering [13] illustrate the breadth of RAG applications. These advances strengthen factual grounding, but none of them explicitly incorporates validation mechanisms that monitor adequacy and completeness.

### 2.3 Corrective and Adaptive Retrieval

Corrective and adaptive retrieval approaches show growing awareness of this gap. Corrective Retrieval Augmentation (CRA) [14] integrates error signals into retrieval, and Self-RAG [15] combines retrieval, generation, and reflection in a unified loop. Adaptive retrieval methods [16] explore query reformulation and contextual retrieval to minimize drift. All share conceptual ground with boosting in that they iteratively improve results. However, their focus remains on generation rather than on dedicated validation.

### 2.4 Learning to Rank

Learning-to-rank methods contribute another relevant dimension. LambdaMART [17] and listwise approaches [18, 19] provide effective techniques for ranking retrieval candidates, while large-scale challenges such as the Yahoo! Learning to Rank dataset [20] established benchmarks for progress.

These methods optimize retrieval quality but do not address the broader question of whether retrieved evidence is sufficient to validate generated answers.

## 2.5 Evaluation and Hallucinations

Finally, evaluation of hallucinations and factuality in natural language generation has gained increasing attention. Ji et al. surveyed hallucination phenomena [1], while Maynez et al. [21] and Zhao et al. [22] analyzed factuality in summarization and question answering. Classical metrics such as precision, recall, and nDCG [23] remain standard, yet they rely on strict string matching and often underestimate semantic adequacy. Surveys of retrieval-augmented methods [24–26] and benchmarks like BEIR [27] provide useful overviews, but none establish explicit validation loops. Recent initiatives such as FEVER [28] and Izacard et al.'s active retrieval paradigm [29] further underline the need for iterative, validation-oriented approaches.

Taken together, the literature reveals three key insights. Boosting highlights the power of iterative error reduction, retrieval-augmented models enhance factual grounding, and evaluation research exposes the limitations of current metrics. What is still missing is an integrated framework that connects these strands by validating retrieval adequacy through iterative mechanisms. Closing this gap is the objective of the methodology described in the following section.

# 3 Methodology

The goal of this work is to develop a validation framework for Retrieval-Augmented Generation (RAG) that integrates principles from boosting and FLARE. Unlike prior research that primarily optimizes retrieval or generation, our focus is on the evaluation loop itself: determining whether retrieved evidence is sufficient, identifying residual gaps, and deciding when expansion is necessary. The methodology is designed to be dataset-agnostic and can be applied to any structured knowledge base. In this section we describe the design principles, system architecture, graph representation, evaluator logic, and performance indicators before introducing the validation scenario in Section 4.

## 3.1 Design Principles

As discussed in Section 2, three strands of research motivate our design: boosting demonstrates the power of iterative error reduction [3, 4, 6], retrieval-augmented models such as RAG, REALM, and FLARE improve factual grounding [2, 8, 5], and evaluation studies expose the limitations of current metrics [1, 21]. From boosting we adopt the idea of residual tracking: in each step, what remains uncovered is treated as error to be addressed. From FLARE we adopt forward-looking expansion: when residuals exceed a threshold, additional retrieval is triggered. Together, these principles transform validation into an iterative process rather than a static one-time assessment.

## 3.2 System Architecture

The framework is organized into four stages. First, the *ingest stage* prepares structured input and artifacts. Second, the *graph construction stage* initializes a knowledge graph that captures elements and relations in a compact form. Third, the *retriever stage* combines sparse retrieval (BM25) with dense embeddings for semantic similarity, similar to approaches in open-domain QA [9]. Finally, the *evaluator stage* applies the boosting- and FLARE-inspired logic that distinguishes our approach from existing retrieval systems.

## 3.3 Graph Representation

Knowledge is represented as a graph to enable transparency and incremental updates. Nodes correspond to citations, documents, or categories, while edges capture references, group membership, or hierarchical relations, as is common in knowledge graph construction [? ? ]. The initial graph is deliberately small, containing only citations and linked documents. Expansion introduces categories or additional documents as new nodes, increasing search space and recall. By tracking which nodes have been covered, the graph directly supports boosting-style residual measurement.

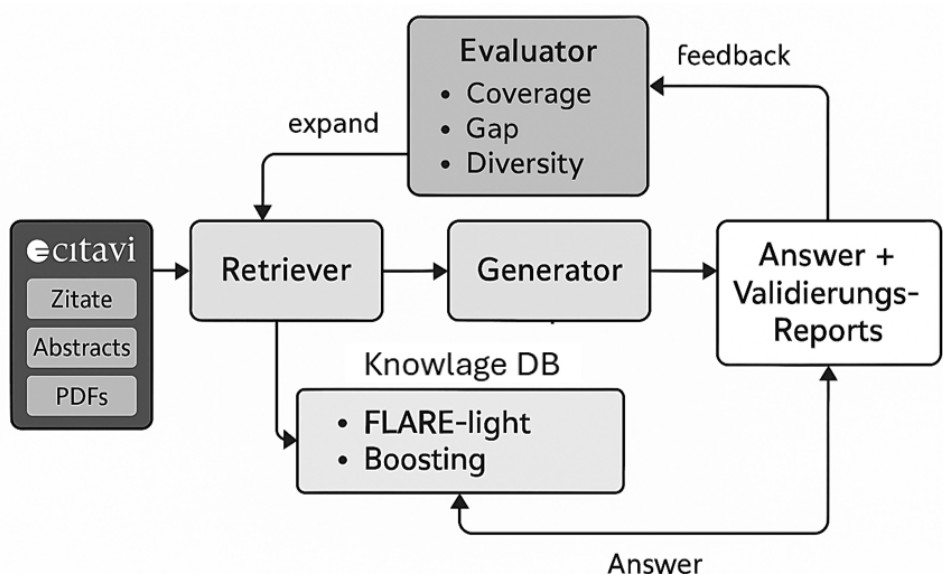

Figure 1: Workflow of the validation framework with Citavi input, RAG retrieval, and FLARE/boosting-inspired evaluation. The dataset-specific component (Citavi) is applied only in the validation scenario described in Section 4.

## 3.4 Evaluator Logic

The evaluator is the methodological core. Inspired by boosting, it calculates residuals by measuring the gap between retrieved evidence and gold references. Inspired by FLARE [5], it then decides whether to expand or stop: if the gap is large, the graph is expanded and retrieval is repeated; if coverage is sufficient, the loop halts. Related approaches such as Corrective Retrieval Augmentation (CRA) [14] and Self-RAG [15] share elements of this idea, but they focus on improving generation rather than providing a dedicated validation loop. Our evaluator reframes these principles as an explicit mechanism for adequacy checking.

## 3.5 Key Performance Indicators

Several key performance indicators operationalize validation. Coverage@k measures whether gold evidence appears among the top-$k$ retrieved items. Because strict matching is often too rigid, we extend this to *Semantic Coverage*, which uses cosine similarity in embedding space. Normalized Discounted Cumulative Gain (nDCG) [23] captures ranking quality and is widely applied in retrieval evaluation [27]. Two additional metrics extend FLARE principles: *Gap-FLARE* quantifies the proportion of uncovered evidence that should trigger expansion, and *Diversity-FLARE* measures the variance among retrieved results to avoid redundancy. Together, these indicators provide a multidimensional perspective on validation that goes beyond classical IR metrics.

## 3.6 Abstraction and Outlook

A crucial property of the methodology is that it remains independent of the specific dataset. It can be applied to enterprise document collections, scientific repositories, or any other structured corpus. In this paper, we use Citavi only as a controlled testbed to examine feasibility, not as part of the method itself. The overall workflow is illustrated in Figure 1, which also anticipates the next section where Citavi is introduced as the validation scenario.

# 4 Validation Scenario

The proposed methodology was validated in a controlled pilot study. The goal of this validation was not to achieve competitive performance, but to demonstrate the feasibility of boosting-inspired

evaluation in a structured environment. This section describes the scope and constants of the experiment, the setup of the validation run, the role of Citavi as a structured testbed, the obtained results, and their interpretation.

## 4.1 Scope and Constants of the Validation

The validation was deliberately constrained in order to focus on the core question of feasibility. Several restrictions were imposed: iterative graph updates were disabled, the number of queries was limited to five, and user feedback was excluded. These choices reduced complexity and ensured that the experiment could be reproduced reliably.

Certain aspects of the setup were treated as constants. The graph was limited to citations and documents, excluding higher-level categories. Citation types in Citavi served as proxy labels, which avoided manual annotation but introduced rigidity. Retrieval was fixed to a combination of BM25 and embedding similarity. Together, these constants provided a stable environment, even though they also introduced biases.

Within this controlled setting, the element under validation was the evaluator. The experiment was designed to test whether boosting-inspired residual tracking and FLARE-style expand/stop logic could be operationalized in practice. The consistent decisions made by the evaluator serve as evidence of feasibility, even if the metrics themselves reveal limitations.

## 4.2 Experimental Setup

The validation run was implemented as a snapshot experiment. At initialization, a small graph was constructed containing citations and their associated documents. Retrieval was carried out using BM25 and dense embeddings, with the two lists merged before evaluation. The evaluator then computed Coverage@5, Semantic Coverage, nDCG@5, Gap-FLARE, and Diversity-FLARE. Logs, result CSV files, and summary JSON files were generated to provide full transparency of the run. In total, five queries were executed, each paired with a gold citation to serve as reference evidence.

## 4.3 Citavi as Structured Testbed

Citavi was chosen as the validation environment because of its structured organization of knowledge. Citations, abstracts, and full-text PDFs are stored in a unified project file, with categories and groups providing additional hierarchical structure. These features map naturally onto graph representations: citations and documents become nodes, while references and categories form edges. Furthermore, citation types (direct quote, summary, paraphrase) function as proxy labels for relevance, allowing evaluation without manual labeling. This makes Citavi an effective testbed for feasibility studies, even though it is not part of the methodology itself.

## 4.4 Results

The outcomes of the validation run are summarized in Table 1. Exact string matching yielded no correct hits, while semantic inspection revealed partial correctness in one case. In all cases, the evaluator returned the decision to expand.

## 4.5 Interpretation

The validation run shows that the evaluator operated consistently and as designed. All five queries resulted in expand decisions, reflecting the detection of residual gaps. Coverage@5 remained at zero under strict string matching, while manual semantic inspection indicated partial adequacy in at least one case. The gap between exact and semantic coverage demonstrates a limitation of the applied metrics. These findings establish the technical feasibility of the evaluation loop and provide the empirical basis for the broader discussion in Section 5.

Table 1: Validation setup, key performance indicators, and results. Gold labels are citations from the Citavi project. Coverage is reported for exact and semantic matching.

| Query | Gold Label | Exact Cov.@5 | Sem. Cov.@5 | nDCG@5 |
|---|---|---|---|---|
| What is FLARE? | FLARE iteratively uses a prediction | 0 | 1 | 0.0 |
| How does RAG combine memory? | RAG combines parametric memory | 0 | 0 | 0.0 |
| What is Gradient Boosting? | Gradient boosting is a generalization | 0 | 0 | 0.0 |
| What is LambdaMART? | LambdaMART combines gradient boosting | 0 | 0 | 0.0 |
| What does REALM interleave? | REALM interleaves knowledge retrieval | 0 | 0 | 0.0 |

## 5    Discussion and Future Work

The validation presented in Section 4 provides a narrow but informative demonstration of the framework. In this section, we move beyond the specific scenario and discuss what the results imply for the methodology introduced in Section 3 and for the broader research gap identified in Section 2.

### 5.1    Implications for the Methodology (Section 3)

The validation confirmed that two central design elements of the methodology are operational: boosting-inspired residual tracking and FLARE-style expand/stop decisions. These findings support the feasibility of treating adequacy as a residual and of embedding expansion as a control mechanism in validation. At the same time, the scope of the experiment revealed which aspects of the methodology remain untested. Iterative updates, semantic coverage metrics, and richer graph representations were not exercised in the pilot run. Their absence does not invalidate the design, but highlights the areas where further empirical work is required. The validation therefore partially substantiates the methodology, while pointing to open components.

### 5.2    Connection to the Research Gap (Section 2)

The limitations observed in Section 4 resonate with prior critiques in the literature. Classical metrics such as Coverage and nDCG underestimated semantic adequacy, echoing findings from hallucination and factuality research [1, 21]. Benchmarks such as BEIR [27] have already called for richer evaluation, but they lack an explicit validation loop. Our framework contributes in this direction by treating validation as an iterative process, informed by residuals and expansion. While the Citavi pilot is minimal, it illustrates that the research gap identified in Section 2 can be addressed with a concrete operational design.

### 5.3    Limitations of the Present Study

The present study is constrained by deliberate design choices: a small number of queries, reliance on proxy labels, and the exclusion of iteration and user feedback. These restrictions were necessary to ensure reproducibility in a proof-of-concept, but they limit the generalizability of the results. The implication is not that the methodology is invalid, but that further studies are required to evaluate its robustness in larger and more diverse settings.

## 6    Conclusion

### 6.1    Summary

This paper proposed a validation framework for Retrieval-Augmented Generation (RAG) that integrates boosting-inspired residual tracking with FLARE-style expand/stop logic. The methodology

shifts the focus from optimizing retrieval or generation to validating adequacy itself, treating uncovered evidence as residuals and using expansion as a control mechanism.

A pilot validation in a Citavi-based testbed confirmed technical feasibility. The evaluator consistently identified residual gaps and triggered expand decisions, demonstrating that the two guiding principles of the methodology can be implemented in practice. At the same time, the restricted scope—five queries, proxy labels, no iterative updates—revealed limitations: classical string-based metrics such as Coverage@k and nDCG underestimated semantic adequacy, and expand decisions could not influence retrieval outcomes. These findings establish a foundation for iterative, feedback-driven validation but stop short of a full performance benchmark.

## 6.2 Future Work

Future work will extend the framework along several directions. First, iterative cycles must be enabled so that residuals and expansion interact dynamically across multiple retrieval rounds. Second, semantic similarity measures will be integrated to capture adequacy beyond surface-level matching, ensuring that paraphrases and equivalent formulations are recognized. Third, richer graph structures should be employed, incorporating categories and cross-document relations to broaden coverage. Fourth, user feedback can be leveraged as an additional residual signal, bridging automated evaluation with practical relevance. Finally, the framework should be applied to larger and more diverse benchmarks such as BEIR as well as to industrial document collections, to assess robustness and scalability.

Taken together, these steps will move the approach from a controlled proof-of-concept toward a practical methodology for improving the reliability of retrieval-augmented generation in scientific and industrial contexts.

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

## A  Technical Appendices and Supplementary Material

Technical appendices with additional results, figures, graphs and proofs may be submitted with the paper submission before the full submission deadline, or as a separate PDF in the ZIP file below before the supplementary material deadline. There is no page limit for the technical appendices.

## Agents4Science AI Involvement Checklist

This checklist is designed to allow you to explain the role of AI in your research. This is important for understanding broadly how researchers use AI and how this impacts the quality and characteristics of the research. **Do not remove the checklist! Papers not including the checklist will be desk rejected.** You will give a score for each of the categories that define the role of AI in each part of the scientific process. The scores are as follows:

- **[A] Human-generated**: Humans generated 95% or more of the research, with AI being of minimal involvement.
- **[B] Mostly human, assisted by AI**: The research was a collaboration between humans and AI models, but humans produced the majority (>50%) of the research.
- **[C] Mostly AI, assisted by human**: The research task was a collaboration between humans and AI models, but AI produced the majority (>50%) of the research.
- **[D] AI-generated**: AI performed over 95% of the research. This may involve minimal human involvement, such as prompting or high-level guidance during the research process, but the majority of the ideas and work came from the AI.

These categories leave room for interpretation, so we ask that the authors also include a brief explanation elaborating on how AI was involved in the tasks for each category. Please keep your explanation to less than 150 words.

IMPORTANT, please:

- **Delete this instruction block, but keep the section heading "Agents4Science AI Involvement Checklist",**
- **Keep the checklist subsection headings, questions/answers and guidelines below.**
- **Do not modify the questions and only use the provided macros for your answers.**

1. **Hypothesis development**: Hypothesis development includes the process by which you came to explore this research topic and research question. This can involve the background research performed by either researchers or by AI. This can also involve whether the idea was proposed by researchers or by AI.

   Answer: **[B]**

   Explanation: The central idea – combining boosting-inspired validation with RAG and using Citavi as the structured testbed – originated from the human researcher. Generative AI contributed by exploring alternative framings and drafting formulations, but it required significant clarification and supervision before aligning with the intended approach. Thus, hypothesis development was primarily human-driven, with AI providing supportive input.

2. **Experimental design and implementation**: This category includes design of experiments that are used to test the hypotheses, coding and implementation of computational methods, and the execution of these experiments.

   Answer: **[C]**

   Explanation: The experimental design and implementation were generated mostly by AI. Generative AI produced the Docker Compose setup, Python application code, and evaluator logic for the validation loop. The human researcher supervised, ensured executability, and applied minimal adjustments (e.g., correct handling of SQLite rows from Citavi and path alignment). Thus, while the technical foundation came from AI, the human role was critical for validation and final operability.

3. **Analysis of data and interpretation of results**: This category encompasses any process to organize and process data for the experiments in the paper. It also includes interpretations of the results of the study.

   Answer: **[D]**

   Explanation: The analysis of data and the interpretation of the results were carried out exclusively by the generative AI. The AI processed the outputs of the validation experiments, produced explanations, and articulated the interpretation of adequacy and gaps in coverage.

The human researcher did not perform independent analysis but only supervised the process from a meta-level. Thus, data analysis and interpretation were exclusively AI-driven.

4. **Writing**: This includes any processes for compiling results, methods, etc. into the final paper form. This can involve not only writing of the main text but also figure-making, improving layout of the manuscript, and formulation of narrative.

   Answer: [D]

   Explanation: The entire writing process—including generation of text, creation of figures and tables, and compilation of the reference list—was carried out exclusively by the generative AI. The human researcher did not contribute to the manuscript text itself but acted only in a supervisory role. Thus, the writing of the paper was exclusively AI-driven.

5. **Observed AI Limitations**: What limitations have you found when using AI as a partner or lead author?

   Description: The observed limitations of generative AI in this project were varied, but the documentation also allows the reader to grasp them for themselves. The link to Github with the entire chat history with ChatGPT, as well as the validation scenario and the paper, is published in the acknowledgements for this paper. Because this chat history already showed a strong trend toward a scientific feasibility review of a concept before the call was launched, it was logical to make it available to your project. The following points from the conversation were particularly noticeable: On the one hand, ChatGPT had difficulty consistently reusing information throughout the entire workflow. For example, towards the end, results from the beginning of the chat history were hardly considered during the paper creation process. ChatGPT frequently relied on its pre-trained background knowledge instead of using the provided project files. Only after explicit inquiries did ChatGPT indicate that it would only superficially review the file. Particularly with papers used for training and also considered in this project, it was impossible to deviate from existing knowledge. Taking the direct approach without critically questioning assumptions, even though critical doubt and methodological rigor are essential in scientific work, was the greatest difficulty in this project. Towards the end of the process, repeated inquiries from the human supervisor were necessary to ensure that the AI had partially considered the provided information and integrated it into the paper. Post-correction for the paper was deliberately omitted. The overall quality of the results was complete and assessable as an independent work with strong support for, for example, an academic paper, but was more reminiscent of a satisfactory bachelor's thesis grade 3.0, as it severely lacked depth, consistency, and critical reflection. This was also influenced by the special setup: the entire workflow was carried out in a single chat, from the brief idea of a term, its contextualization, deriving possible synergies, identifying the use case and research question, and finally creating the paper itself. In cases where individual steps are examined over several sessions, the process of creating a scientific paper was deliberately followed from start to finish in a continuous dialogue. This structure created a workflow similar to supervised student work: The human took on the role of supervisor, while the AI took on the role of the students and was guided to do scientific work. The AI was able to create depth for clearly defined sub-goals, but often lost the overall overview and rushed into creating final versions, which is why several loops were created. The ChatGPT fluctuated between superficial overviews and repeated refinements of simple to-dos with limited added value. These dynamics—including strengths and weaknesses—are documented in the chat transcript included in the repository formatted for readability.

